# Is Being Physically Active Enough or Do People with Parkinson’s Disease Need Structured Supervised Exercise? Lessons Learned from COVID-19

**DOI:** 10.3390/ijerph19042396

**Published:** 2022-02-19

**Authors:** Josefa Domingos, Carlos Família, Júlio Belo Fernandes, John Dean, Catarina Godinho

**Affiliations:** 1Grupo de Patologia Médica, Nutrição e Exercício Clínico (PaMNEC) do Centro de Investigação Interdisciplinar Egas Moniz (CiiEM), Monte de Caparica, 2829-511 Caparica, Portugal; domingosjosefa@gmail.com (J.D.); carlosfamilia@egasmoniz.edu.pt (C.F.); jfernandes@egasmoniz.edu.pt (J.B.F.); 2Department of Neurology, Radboud University Medical Center, Donders Institute for Brain, Cognition and Behaviour, 6500 HB Nijmegen, The Netherlands; 3Triad Health AI, Aurora, CO 80012, USA; john@johnmdean.com; 4Molecular Pathology and Forensic Biochemistry Laboratory (MPFBL), Centro de Investigação Interdisciplinar Egas Moniz (CiiEM), Monte de Caparica, 2829-511 Caparica, Portugal

**Keywords:** social isolation, Parkinson’s disease, exercise, sedentary behaviour, active behaviour

## Abstract

Social isolation imposed by the COVID-19 pandemic negatively impacted people’s lifestyles and daily activities. In this work we compared pre- and post-pandemic clinical outcomes in people with Parkinson’s disease, to assess differences according to the type of behaviour and exercise habits adopted by participants. After two months of COVID-19 lockdown, we assessed: changes in exercise behaviour; motor and non-motor aspects of daily life experiences (MDS-UPDRS I & II); activities of daily living (The Schwab & England scale); quality of life (Parkinson Disease Questionnaire); sleep (Parkinson Disease Sleep Scale); falls; and Clinical Global Impression Change. Twenty-seven individuals aged between 57 and 92 years old participated; from these, ten individuals (37%) completely interrupted physical activities, while seventeen (63%) maintained some level of active lifestyle. Regardless of whether they remained active or not, all participants perceived a significant worsening of their clinical condition, reporting an increase in difficulties completing daily activities or chores (37%) and worsening of their health condition (51.8%). The quantifiable influence of exercise habits was borderline for the group who kept active. The active group seem to have a better self-perception of their health condition, although it was not enough to show a clear benefit. People with Parkinson’s disease should be informed that being physically active may not be enough and more structured exercise could be needed.

## 1. Introduction

The social isolation imposed by the novel coronavirus 2 (COVID-19 pandemic) caused a significant negative impact on people’s lifestyles and daily activities around the world [1]. Health care systems internationally rapidly implemented infection and environmental control strategies to control COVID-19. While quarantine was considered a highly-efficacious, widely-implemented strategy to minimize exposure risk and reduce the spread of the disease [2], it imposed severe sudden changes in daily routines. For people with Parkinson’s disease (PD), these changes led to reduced physical activity and worsening social isolation, resulting in several negative impacts upon symptoms of the disease in addition to lower overall health status [3].

Even before COVID restrictions were imposed, people with PD are on average 29% less physically active than age-matched healthy controls, particularly in the presence of greater disease severity, gait impairment and more disability in daily living [3,4]. Physiotherapy and exercise can improve physical functioning [5,6,7,8]. However, continuous, ongoing exercise is needed to maintain results and promote long-term optimal levels of physical activity [9].

As social restrictions imposed by the COVID-19 pandemic began to take place, people with PD could have been reluctant or unable to make changes in their routines. In addition, patients are often unaware of the severity and impact of inactivity on their symptoms, and compliance with maintaining exercise remains a critical challenge [10,11]. A recent study showed that physical activity was strongly associated with improved well-being outcomes when compared to inactive individuals during COVID-19 lockdown [12]. Key questions arise about what health-promoting measures are needed to encourage an active lifestyle and ensure a continuum of care during social isolation periods.

People with PD can benefit from mild exercise habits like self-guided brisk walking [13] or from somewhat more moderate-intensity physical activity like Nordic walking [14], as well as from more planned structured exercise-based interventions [4,5]. Importantly, studies have shown how structured exercise such as high-intensity treadmill exercise [15] or bicycle aerobic exercise [16] can affect PD symptoms. Improving our understanding of the need for more structured exercise, or of just being physically active, in people with PD will provide valuable clinical information on how to prescribe and maximise the long-term therapeutic benefits of exercise in this population. Targeting specific movement aspects in PD can be expected to improve perceived difficulties in daily life activities. Knowledge of the effects of one type of exercise or another should inform clinical decision-making and recommendations given to patients. Ultimately, is keeping physically active alone enough exercise for a person with PD? Thus, people with Parkinson’s disease and healthcare and exercise professionals involved in Parkinson’s disease management should be able to benefit from information about physical exercise prescription.

Here we aim to answer the question: Is being physically active enough or do people with Parkinson’s disease need structured exercise? We did this by comparing clinical outcomes from before the pandemic lockdown with the same clinical measures following 2 months of isolation in people with PD to assess differences in motor and non-motor aspects of daily life experiences, quality of life, sleep patterns, and falls, in comparison with patients’ perceived worsening after these two months of COVID-19 lockdown according to the type of exercise behaviour adopted.

## 2. Methods

### 2.1. Design

This is a longitudinal prospective study with a pre- and post-study design. Our goal was to assess changes in several measures of disability, independence in activities of daily living (ADL), motor and non-motor symptoms, quality of sleep, self-perception of disease severity, and occurrence of falls after two months of social isolation associated with the COVID-19 lockdown, according to type of exercise behaviour.

### 2.2. Sampling and Recruitment

The sampling method selection was non-probabilistic by convenience. Participants were invited and included if they were participating in physiotherapy immediately before the pandemic lockdown restrictions in Portugal. All participants were invited by telephone to participate in this study. Recruitment was performed in March 2020.

### 2.3. Data Collection

Patients were assessed when care was initially interrupted (March 2020) and two months later via telephone interviews and email exchanges. Disease severity (Hoehn and Yahr—H&Y) was assessed based on patient records. A number of assessment tools were sent by email to be filled in by patients or caregivers, including the following: Motor Aspects of Daily Life Experiences (MDS-UPDRS II) scored from “0” to “52”, where the higher value reflects greater severity of symptoms; and Quality of life (Parkinson Disease Questionnaire PDQ-8). For this scale, the qualitative output was assigned quantitative values in order to make these results quantifiable. So, when the answers were “never”, the value “0” was assigned, for the answers “Occasionally” the value “1”, for “Sometimes” the value “2”, for “Often” the value “3” and finally for “Always” or “cannot do at all”, the value “4”. The lower the final sum, the better the quality of life perceived by the individuals [17], Sleep and nocturnal disability were assessed using a Portuguese version of the original 15-item Parkinson’s Disease Sleep Scale (PDSS-2) on which individual items are scored from “0” to 10, with the total score ranging from 0 to 150, where higher scores indicate greater impairment [18]. Other clinical data were collected by telephone interview, included Non-Motor Aspects of Daily Life Experiences Unified Parkinson’s Disease Rating Scale (MDS-UPDRS- I) scored from “0” to “52”, where the higher value reflects greater severity of symptoms [19]; Clinical Global Improvement or Change (CGIC), applied only during post-assessment; answer options 1 = very much improved; 2 = much improved; 3 = minimally improved; 4 = moderately ill; 5 = no change; 6 = minimally worse; 7 = much worse; 8 = very much worse [20]; and Activities of Daily Living (The Schwab & England ADL scale—S&E) scored from 0% = completely dependent to 100% = completely independent for ADL [21]. Occurrence of falls was also recorded. Each participant kept a monthly falls diary. The overall number of falls per participant during the study period was analyzed.

### 2.4. Data Analyses

The statistical analysis of the questionnaire data was performed using the R language and environment for statistical computing v.4.1.2 [22], using RStudio integrated development environment v.2021.09.0 [22].

Descriptive statistic measures of count, mean, standard deviation, median, minimum, maximum and range were computed for sample characterisation, using the function table1 from the table1 v.1.4.2 library [22] for R, and for the scores obtained for each questionnaire grouped by moment (pre- and post-social isolation) and by patient exercise activity (i.e., whether it was maintained or interrupted), using the “describe By” function from the psych v.2.1.9 library [22] for R. All plots were generated using the ggplot2 library [23] for R.

Linear mixed-effects models were used to analyse the effect of the assessment moment and exercise interruption, as well as the interaction between them on the score of MDS-UPDRS I, MDS-UPDRS II, PDQ-8, PDSS2, S&E, with random intercepts per individual. Factorial analysis of variance (ANOVA) of type II with Satterthwaite’s approximation for degrees of freedom was subsequently performed to evaluate the significance of these effects, using the “ANOVA” function from the stats library v.4.1.2 [22] for R. All models were defined using the “lmer” function from the lmerTest library v.3.1.3 [24] for R and estimations were performed using the restricted maximum likelihood algorithm (RMEL). Effect sizes of partial η^2^ were computed with the function F_to_eta2 from the effect size v.0.6.0.1 library [25] for R. The normality and homoscedasticity of the residuals were evaluated through the analysis of the residual plot and quantile–quantile of the residuals as a function of the predicted values, of the index plot of the residues as a function of the subject index, and of the histogram of the residuals, all obtained using the “resid_panel” function from the ggResidpanel v.0.3.0 library [26]. Wilcoxon rank sum test with continuity correction was used to evaluate the effect of exercise interruption in the CGIC score.

### 2.5. Ethics and Procedures

This study follows the principles of the Declaration of Helsinki. Before conducting this study, a research protocol was approved by the Board of Directors and the Ethical Committee of Egas Moniz—Cooperativa de Ensino Superior, CRL (Protocol ID 892/2020).

All participants had given informed consent to participate in the study before any data collection.

## 3. Results

### 3.1. Participants

Twenty-seven individuals between 57 and 92 years old (M = 71.93, SD = 7.69 years old) participated in the study. The majority of participants were male (22; 81.49%), retired (24; 88.89%), and married (19; 70.37%). Most participants had a diagnosis of Idiopathic Parkinson’s (23; 85.19%) with a mean of 7.07 ± 4.12 (SD) years after the diagnosis, and were in Hoehn and Yahr (H&Y) Stage III (15; 55.56%). Relevant patient demographic and disease severity (H&Y) can be seen in Table 1.

### 3.2. Exercise’s Habits/Lifestyle Behaviour

Ten individuals (37%) reported that they had completely interrupted exercise or physical activity behaviours. Seventeen (63%) had maintained elements of active lifestyle, including informal exercise routines such as daily outdoor walks, home exercises and online sections. On average, those who maintained an active lifestyle were younger and had a lower degree of disease severity (H&Y).

In Table 2 we can observe the total scores of clinical scales for both groups with PD.

### 3.3. Non-Motor Symptoms (MDS-UPDRS Part I)

A linear mixed-effects model was used to evaluate the effects of assessment moment (pre- and post-social isolation), exercise habits (maintained and interrupted), and the interaction between assessment moment and exercise habits on the score of MDS-UPDRS Part I (Figure 1a). A random intercept was defined per individual to accommodate repeated measures. Subsequent type II analysis of variance (ANOVA) with Satterthwaite’s approximation for degrees of freedom shows that the effect of the timing of the assessment (β = 3.06, SE = 0.64) influences the model significantly (F(25) = 47.42, *p*-value < 0.01, partial η^2^ = 0.65), and that the effect of exercise habits (β = 1.34, SE = 2.44) and of the interaction between the timing of the assessment and exercise habits (β = 1.14, SE = 1.05) do not significantly influence the model (F(25) = 0.64, *p*-value = 0.43, partial η^2^ = 0.02 and F(25) = 1.19, *p*-value = 0.29, partial η^2^ = 0.05, respectively) for a significance level of 0.05. Analysis of the residuals plot, Q–Q plot, Index plot and Histogram shows no clear violations to the assumptions of normality and homoscedasticity of the residuals for this model.

The evaluation of cognition, behaviour and mood measured by MDS-UPDRS Part I (Figure 1a and Table 1) showed significant statistical differences between pre- and post-assessment periods in general, with a large effect size (*p*-value < 0.01, partial η^2^ = 0.65). These individuals perceived a significant worsening of their clinical condition related to this non-motor symptomatology during this period of social isolation, regardless of whether they remained minimally active or not. However, through the observation of Figure 1a, we can see a worse symptomatology (higher values) for individuals who became inactive.

### 3.4. Motor Symptoms (MDS-UPDRS Part II)

A linear mixed-effects model was used to evaluate the effects of the timing of the assessment, exercise habits, and of the interaction between them on the score of MDS-UPDRS Part II, with a random intercept per individual (Figure 1b). Subsequent type II analysis of variance with Satterthwaite’s approximation for degrees of freedom shows that the effect of the timing of the assessment (β = 2.47, SE = 0.61) affects the model significantly (F(25) = 35.51, *p*-value < 0.01, partial η^2^ = 0.59), and that the effect of exercise habits (β = 2.28, SE = 3.88) and of the interaction between the timing of the assessment and exercise habits (β = 1.13, SE = 1.00) do not significantly influence the model (F(25) = 0.55, *p*-value = 0.47, partial η^2^ = 0.02 and F(25) = 1.27, *p*-value = 0.27, partial η^2^ = 0.05, respectively) for a significance level of 0.05. Analysis of the residuals plot, Q–Q plot, Index plot and Histogram shows no violations to the assumptions of normality and homoscedasticity of the residuals for this model.

The evaluation of motor aspects in experiences of daily living measured by MDS-UPDRS Part II (Figure 1b and Table 1) showed that all final clinical motor outcomes were significantly worsened and with a large effect size (*p*-value < 0.01, partial η^2^ = 0.59) during social isolation, without distinguishing any effect related to exercise habits (*p*-value = 0.27), although we can observe from Figure 1b a worse clinical condition (higher values) for individuals who became inactive (interrupted).

### 3.5. Quality of Life

A linear mixed-effects model was used to evaluate the effects of the timing of the assessment, exercise habits and interaction between them on the score of PDQ-8, with a random intercept per individual (Figure 1c). Subsequent type II analysis of variance with Satterthwaite’s approximation for degrees of freedom shows that the effect of the timing of the assessment (β = 1.35, SE = 0.44) affects the model significantly (F(25) = 23.90, *p*-value < 0.01, partial η^2^ = 0.49), and that the effect of exercise habits (β = −1.68, SE = 2.34) and of the interaction between the timing of the assessment and exercise habits (β = 0.95, SE = 0.72) do not influence the model significantly (F(25) = 0.27, *p*-value = 0.61, partial η^2^ = 0.01 and F(25) = 1.72, *p*-value = 0.20, partial η^2^ = 0.06, respectively). Analysis of the residuals plot, Q–Q plot, Index plot and Histogram shows no violations to the assumptions of normality and homoscedasticity of the residuals for this model.

The evaluation of quality of life measured by PDQ-8 (Figure 1c and Table 1) showed that the quality of life for these patients worsened significantly post-social isolation, with a large effect size (*p*-value < 0.01, partial η^2^ = 0.49), again, without distinguishing any effect related to exercise habits (*p*-value = 0.20).

### 3.6. Disability and Independence in Activities of Daily Living (ADL)

A linear mixed-effects model was used to analyse the effects of the timing of the assessment, exercise habits and of the interaction between them, on the score of S&E scale, with a random intercept per individual (Figure 1d). Subsequent type II analysis of variance with Satterthwaite’s approximation for degrees of freedom shows that the effect of the the timing of the assessment (β = −2.35, SE = 1.13) affects the model significantly (F(25) = 16.96, *p*-value < 0.01, partial η^2^ = 0.40), and that the effect of the timing of the assessment (β = −4.12, SE = 8.23) and of the interaction between the timing of the assessment and exercise habits (β = −3.65, SE = 1.86) do not influence the model significantly (F(25) = 0.53, *p*-value = 0.47, partial η^2^ = 0.02 and F(25) = 3.84, *p*-value = 0.06, partial η^2^ = 0.13, respectively). Analysis of the residuals plot, Q–Q plot, Index plot and Histogram shows a small deviation to normality and no violations to the assumption of homoscedasticity of the residuals for this model.

The difficulties that patients had in completing daily activities or chores and changes during the social isolation period assessed using Schwab and England Scale (S&E scale) showed significant statistical differences with a large effect size (*p*-value < 0.01, partial η^2^ = 0.40) between the pre- and post-assessments periods for the group who had completely interrupted exercise habits (Figure 1d). Although the analysis concerning the interaction between the timing of the assessment and exercise habits does not show a significant difference for the influence of exercise habits, we must emphasize that the value obtained for the statistical significance is borderline (*p*-value = 0.06).

### 3.7. Sleep Assessment

A linear mixed-effects model was used to analyse the effects of the timing of the assessment, exercise habits, and of the interaction between them on the score of PDSS-2, with a random intercept per individual (Figure 1e). Subsequent type II analysis of variance with Satterthwaite’s approximation for degrees of freedom shows that the effect of the timing of the assessment (β = 1.31, SE = 0.70), of exercise interruption (β = −7.68, SE = 8.13) and of the interaction between assessment moment and exercise habits (β = −1.51, SE = 1.13) do not influence the model significantly (F(24) = 1.76, *p*-value = 0.20, partial η^2^ = 0.07; F(24) = 1.08, *p*-value = 0.31, partial η^2^ = 0.04; and F(24) = 1.79, *p*-value = 0.19, partial η^2^ = 0.07, respectively). Analysis of the residuals plot, Q–Q plot, Index plot and Histogram shows no violations to the assumptions of normality and homoscedasticity of the residuals for this model.

Participants’ nocturnal disability and sleep measured by Parkinson’s Disease Sleep Scale (PDSS-2) (Figure 1e and Table 1) showed that there were no significant changes in sleep quality perceived by these patients during this period, independently of the exercise habits adopted.

### 3.8. Falls

Falls occurred in four (14%) patients during the 2 month period. Two of these individuals had no history of previous falls. Three of these participants belong to the group that had interrupted exercise habits and one belongs to the group who had maintained an active lifestyle.

### 3.9. Patient’s Clinical Global Impression Assessment

Patients’ Clinical Global Impression of their disease was only asked for after 2 months of social isolation to assess the participants’ perception of the evolution of their general health status during the lockdown. These results showed inconsistent change during the pandemic period. Fourteen patients (51.85%) reported that they felt “minimally worse” and 13 patients (48.15%) did not report any changes between the periods analysed.

Additional analyses were conducted to explore relationships between perceived worsening, individuals who interrupted all exercise and those who had maintained some exercise habits. Patients who interrupted exercise were shown to have a higher CGIC score (Mdn = 6.0, self-perception of 0“minimally worse”), than those who maintained exercise habits (Mdn = 5.0, self-perception of “no change”). A Wilcoxon signed-rank test indicated that this difference was not statistically significant (W = 51, *p*-value = 0.056) for a significance level of 0.05, though this value is also borderline.

## 4. Discussion

As the COVID-19 pandemic continues, the high demand for clinical resources and strategies requires us to rationalize the use of these resources. Knowledge of the clinical impact and the consequences of withholding health care from people with PD will aid health systems in better clinical decisions in order to set up proper prevention and management strategies. We assessed the impact of social isolation and healthcare restrictions in 27 individuals with PD. Overall, irrespective of whether they remained active or not, participants showed a worsening in most outcome measures over a period of two months. Participants reported an increase in difficulties in completing daily activities or chores (37%) and worsening of their health condition (51.8%), but the influence of exercise habits was borderline with a medium effect size for the group who kept active (F(25) = 3.84, *p*-value = 0.06, partial η^2^ = 0.13). Based upon scores from MDS-UPDQRS I and II and S&E scale, the active group seems to have a better maintenance of their health condition, but overall, this was not enough for them to perceive better health status. The maintenance of an active lifestyle does not seem to exert a clear benefit on the patient’s self-perception of health status. For people with PD to maintain their health status, it may not be enough just to keep active, and additional clinical supervision might be needed.

Given the expected continuation of COVID-19 restrictions, there is a need for health care systems to set up proper prevention and management strategies for exercise and activity changes in people with PD. Reinforcing the need for individuals to maintain an active lifestyle is very important for the overall population’s health and even more so for people with PD [3,12]. However, simply telling people to exercise is not enough. Strategies to raise awareness and facilitate physical activity supported by health professionals at home should be implemented expeditiously. Sharing clinical findings of the negative impact with patients allows them to make well-informed decisions about their actions and better motivates them towards specific care at home and in the community. In addition, reinforcing social support online is also vital. As health restrictions due to the pandemic impact common social support, it is imperative to capitalize upon technologies in order to provide the support to keep people motivated to be active and supervised during isolation. This social support and guidance are considered a significant contributor to exercise behaviour in inactive individuals as it enhances exercise self-efficacy and motivation [27], as well as helps bypass several barriers to, and involve facilitators for, exercise behaviour in PD [10,11,28].

Clinical Global Impression of the patients’ disease did not demonstrate clear differences between groups of participants during the pandemic period. Fourteen patients (51.85%) reported that they felt “minimally worse”, and 13 patients (48.15%) did not report any changes between the periods analysed. Notably, however, there were discrete differences between individuals who interrupted all exercise and those that maintained exercise habits. Ten of the individuals who reported feeling worse (37%) were participants who had interrupted all exercise-based activities since the COVID-19 outbreak, while only four. (23.5%) participants who maintained some physical activity level reported that they were worse during the COVID-19 quarantine. Although this did not really achieve statistical significance (*p*-value = 0.056), it does highlight a potential benefit of maintaining exercise, particularly under the supervision of healthcare or exercise professionals. People with Parkinson’s patients will benefit from clear guidance on which exercise and physical activities are most beneficial, how often to do them, which is a safe intensity, how many resting periods are needed, which barriers to expect, etc., [5,29]. However, further investigation is necessary to properly assess these factors.

A reduction in exercise during the COVID-19 pandemic is expected to lead to a worsening of motor symptoms in PD and is in line with previous research [3,12,30]. Non-motor symptoms such as insomnia, sleep, psychological stress or constipation may also worsen due to lack of physical activity/active lifestyle [3]. Interestingly, individuals who did not perceive any deterioration on the “Patient’s Clinical Global Impression” declined 4.3 points on the MDS-UPDRS (I and II). Several studies have highlighted the challenges to people with PD in perceiving difficulties, and suggest that self-perceived insight into the severity of illness may be impaired due to basal ganglia dysfunction [31,32,33]. Often, it is a family member who notices changes and supports monitoring and correcting strategies [34,35]. A more significant difference in clinical outcomes would allow for a clearer perception of worsening by the individual with PD and more education on self-assessment could be valuable.

Related to difficulties completing activities of daily living (ADLs), 37% of participants reported increased difficulty, while 63% identified no change. However, an active lifestyle was shown to exert a positive influence on self-perception of capacity to maintain independence in the activities of daily living as measured by S&E scale.

Interestingly, the group ‘exercise interrupted’ in both pre and post-social isolation perceived a better quality of life and lower impairment in quality of sleep than the exercise maintained group. Most exercise intervention research studies [36,37,38] target people with mild to moderate PD symptoms, so little is known about how advanced stages might benefit, or perceive this benefit. Thus, we hypothesize that the group that interrupted activity was mainly constituted of older people (77.0 to 68.9 years) and worse disease and level of disability (H&Y 3 to 2.5), who may not easily perceive worsening and, thus, this might not impact quality of life assessments. Living with the disease for longer periods may also reflect a better adjustment to disability and potentially hamper self-assessment due to potential cognitive limitations.

We identify some limitations in the study. First, the COVID-19 pandemic provided a unique possibility to study the impact of a lockdown on PD patients. Nonetheless, it also restricted the opportunity to execute a sample size calculation and recruit a matched control group. Second, the unavoidable lack of control over the variability of both the quantity and the quality of exercises instituted during lockdown also complicates a full comparison. Third, the inability to assess MDS-UPDRS III post-COVID-19 limits our ability to test the overall clinical worsening and total scores. Even though it is feasible, but not uncontroversial, to perform this assessment online [39], most of these patients experienced limitations in their use of technology which imposed significant restrictions on online access. Fourth, even though MDS-UPDRS Part I covers several aspects of non-motor symptoms, additional assessment of non-motor and cognitive aspects would have given us additional information. Finally, all of the underlying reasons why participants inactive participants interrupted all exercise were not fully assessed. Identifying the barriers and facilitators to physical activity/active lifestyle [40] and motivation to exercise during the pandemic would allow us to define clear solutions in order to increase adherence to exercise via self-care or telemedicine [41]. Due to these limitations, all results must be interpreted with the necessary caution.

## 5. Conclusions

Our results show that participants were worse in the majority of outcomes assessed during the pre- and post-COVID-19 lockdown period, independent of their exercise habits.

The right initiatives must be implemented to maintain optimal care in the home setting when care cannot be provided directly, such as remote exercise programs or distance guidance exercise for Parkinson’s disease supervised by healthcare or exercise professionals. Healthcare (physiotherapist, occupational therapist, others) and exercise professionals that work with people with Parkinson’s and prescribe exercise to patients will benefit from using this data as an example, in order to raise patient awareness of the effect of one type of exercise over the other.

In the near term, persons with PD should be well-informed that they need to look for specialised intervention to maintain their health status. Merely keeping active is not enough to feel health benefits.

## Figures and Tables

**Figure 1 ijerph-19-02396-f001:**
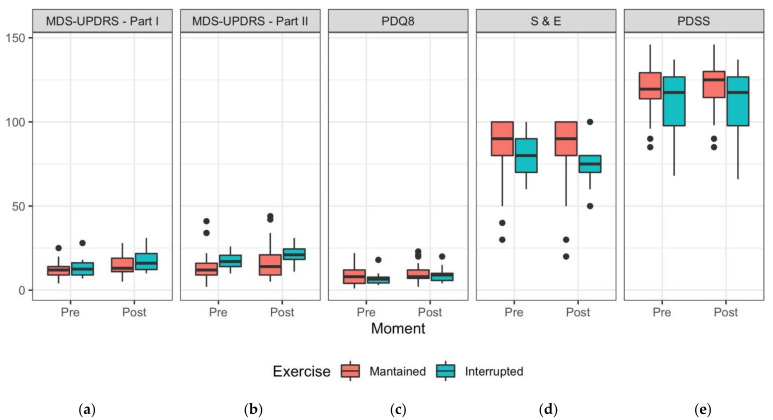
(**a**). Evaluation of cognition, behaviour and mood measured by Movement Disorders Society-Unified Parkinson Disease Ratting Scale Part I (MDS-UPDRS I) for pre- and post-assessment moments in terms of function of exercise habits. (**b**). Evaluation of motor aspects of experiences of daily living measured by Movement Disorders Society-Unified Parkinson Disease Rating Scale Part II (MDS-UPDRS II) for pre- and post-assessment moments in terms of function of exercise habits. (**c**). Evaluation of quality of life measured by PDQ-8 for pre- and post-assessment moments in terms of function of exercise habits. (**d**). Evaluation of disability and independence in activities of daily living measured by Schwab & England ADL scale (S&E) for pre- and post-assessment moments in terms of function of exercise habits. (**e**). Evaluation of nocturnal disability and sleep measured by Parkinson’s Disease Sleep Scale (PDSS) for pre- and post-assessment moments in terms of function of exercise habits.

**Table 1 ijerph-19-02396-t001:** Participant Sociodemographic and Clinical Characteristics.

	Patient’s Exercise		Overall (*N* = 27)
	Maintained (*N* = 17)	Interrupted (*N* = 10)	
Age			
Mean (SD)	68.9 (6.37)	77.0 (7.29)	71.9 (7.69)
Median [Min, Max]	69.0 (57.0, 80.0)	77.5 (66.0, 92.0)	70.0 (57.0, 92.0)
Sex			
Female	4 (23.5%)	1 (10.0%)	5 (18.5%)
Male	13 (76.5%)	9 (90.0%)	22 (81.5%)
Hoehn & Yahr			
Mean (SD)	2.50 (0.866)	3.00 (0.471)	2.69 (0.774)
Median [Min, Max]	2.50 (1.00, 4.00)	3.00 (2.00, 4.00)	3.00 (1.00, 4.00)

**Table 2 ijerph-19-02396-t002:** Clinical characteristics of Parkinson’s disease patients in pre- and post-assessment moments in terms of function of exercise habits.

	Pre-Social Isolation		Post-Social Isolation	
	Exercise Maintained (*N* = 17)	Exercise Interrupted (*N* = 10)	Exercise Maintained (*N* = 17)	Exercise Interrupted (*N* = 10)
**MDS-UPDRS Part I**				
Mean (SD)	12.1 (5.26)	13.4 (6.45)	15.1 (6.24)	17.6 (7.00)
Median (Min, Max)	12.0 (4.00, 25.0)	12.5 (7.00, 28.0)	13.0 (5.00, 28.0)	16.0 (10.0, 31.0)
**MDS-UPDRS Part II**				
Mean (SD)	15.4 (11.0)	17.7 (5.54)	17.9 (11.9)	21.3 (5.46)
Median (Min, Max)	12.0 (2.00, 41.0)	17.0 (10.0, 26.0)	14.0 (5.00, 44.0)	21.0 (11.0, 31.0)
**PDQ8**				
Mean (SD)	16.8 (6.71)	15.2 (4.34)	18.1 (6.41)	17.5 (4.88)
Median (Min, Max)	16.0 (9.00, 30.0)	14.5 (11.0, 26.0)	16.0 (10.0, 31.0)	17.0 (12.0, 28.0)
**S & E**				
Mean (SD)	84.1 (22.4)	80.0 (12.5)	81.8 (25.3)	74.0 (13.5)
Median (Min, Max)	90.0 (30.0, 100)	80.0 (60.0, 100)	90.0 (20.0, 100)	75.0 (50.0, 100)
**PDSS**				
Mean (SD)	119 (17.3)	111 (23.9)	120 (17.4)	111 (24.3)
Median (Min, Max)	120 (85.0, 146)	118 (68.0, 137)	125 (85.0, 146)	118 (66.0, 137)
Missing	1 (5.9%)	0 (0%)	1 (5.9%)	0 (0%)
**CGIC**				
Mean (SD)	NA	NA	5.41 (0.618)	5.80 (0.422)
Median (Min, Max)	NA	NA	5.00 (5.00, 7.00)	6.00 (5.00, 6.00)

Movement Disorders Society-Unified Parkinson Disease Rating Scale Part I and II (MDS-UPDRS I–II); Parkinson Disease Questionnaire-8 (PDQ-8); Schwab & England ADL scale (S&E); Parkinson’s Disease Sleep Scale (PDSS-2); Clinical Global Improvement or Change (CGIC). NA—not applicable.

## Data Availability

The data presented in this study are available on request from the first author.

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
