# Peer review of "Is Being Physically Active Enough or Do People with Parkinson’s Disease Need Structured Supervised Exercise? Lessons Learned from COVID-19"

_ijerph, 2022, doi:10.3390/ijerph19042396_

Round 1

Reviewer 1 Report

This manuscript was designed to investigate the effect of the 2-months COVID19-caused pandemic lockdown on clinical outcomes in people with Parkinson´s disease.  The type of behavior and exercise habits adopted by participants during 2 months of social distancing was also assessed. Although the study sounds interesting, there are limitations that compromise the findings of the study, including the absence of a control group and sample size determination.  

Minor comments
Introduction 
Line 32-34 – I think it's a very short paragraph. 

Methods
Was sample size calculated? Please provide this information

Results
I suggest the authors provide the effect size and discuss them

Author Response

We want to thank the reviewers for their comments and efforts towards improving our manuscript. We have incorporated changes to reflect your suggestions. We have highlighted the changes within the manuscript.

Reviewer 1

This manuscript was designed to investigate the effect of the 2-months COVID19-caused pandemic lockdown on clinical outcomes in people with Parkinson´s disease.  The type of behavior and exercise habits adopted by participants during 2 months of social distancing was also assessed. Although the study sounds interesting, there are limitations that compromise the findings of the study, including the absence of a control group and sample size determination.  

Minor comments:

 Comment 1 - Introduction- Line 32-34 – I think it's a very short paragraph. 

Answer to reviewer: We thank the reviewer for the comments. We agree and given that the topics interconnect we have connected the paragraph and changed the text accordingly.

Comment 2 - Methods - Was sample size calculated? Please provide this information

Answer to reviewer: As the confinement occurred, we invited all possible people with Parkinson's that were undertaking physiotherapy before to participate. We did not calculate sample size. We have placed in the study limitations the following text: “the COVID-19 pandemic provided a unique possibility to study the impact of a lockdown on PD patients. Nonetheless, it also restricted the opportunity to execute a sample size calculation and recruit a matched control group.”

Comment 3 - Results -I suggest the authors provide the effect size and discuss them

Answer to reviewer: We thank the reviewer and agree with the suggestion. We have added the effect size in the text.

Reviewer 2 Report

The present paper is original and well-structured. However, there are some issues that need to be addressed in order to enrich the manuscript prior to publication.

Abstract section

And is interesting if authors answer to the question formulated in the title. Do people with Parkinson need a structured exercise?

It is pertinent to add suggestions to improve health condition of people with Parkinson.

The authors should expose for who these findings will be important.

Introduction section

Can the authors clarify the pertinence of the study?

It is suggested to report what kind of benefits people with Parkinson Disease will obtain from brisk walking or planned structured exercise-based interventions. And should expose what is incorporated in these planned structured exercise-based interventions (i.e., what exercises).

The authors should add the hypothesis of the study.

Discussion section

The authors should expose the importance of the sport sciences professionals and their supervision. How people know if they are doing a proper exercise? With the proper intensity? How many time to rest? Manner of doing the exercise? …

It is needed to support the findings obtained with studies/references. What are provided in the literature? What other authors found?

Conclusion section

It is suggested to clarify the right initiatives to maintain optimal care? Right initiatives are too general.

Can the authors expose some examples for the new approaches to improve healthcare suggested by them in the conclusion section?

The authors mentioned that “keeping active is not enough to feel benefits in health”, but how did they control the routines and exercises made by people with Parkinson Disease? Were people with Parkinson supervised during the active routines?

Author Response

We want to thank the reviewers for their comments and efforts towards improving our manuscript. We have incorporated changes to reflect your suggestions. We have highlighted the changes within the manuscript.

Reviewer 2

The present paper is original and well-structured. However, there are some issues that need to be addressed in order to enrich the manuscript prior to publication.

Comment 1 - Abstract section

And is interesting if authors answer to the question formulated in the title. Do people with Parkinson need a structured exercise? It is pertinent to add suggestions to improve the health condition of people with Parkinson. The authors should expose for who these findings will be important.

Answer to reviewer: We thank the reviewer for the comments. We have added to the abstract the following text:

“People with Parkinson's should be informed that being physically active may not be enough and more structured exercise could be needed.”

Comment 2 - Introduction section

Can the authors clarify the pertinence of the study? It is suggested to report what kind of benefits people with Parkinson Disease will obtain from brisk walking or planned structured exercise-based interventions. And should expose what is incorporated in these planned structured exercise-based interventions (i.e., what exercises).

The authors should add the hypothesis of the study.

Answer to reviewer: To better clarify the pertinence of the study, we have added the following text in the introduction:

“Studies have shown how structured exercise such as high-intensity treadmill exercise [15] or bicycle aerobic exercise [16] can affect PD symptoms. Improving our understanding about the need for more structured exercise or just being physically active in people with PD will provide valuable clinical information on how to prescribe and maximize the long-term therapeutic benefits of exercise in this population.

Comment 3 - Discussion section

The authors should expose the importance of the sport sciences professionals and their supervision. How do people know if they are doing proper exercise? With the proper intensity? How many time to rest? Manner of doing the exercise? …It is needed to support the findings obtained with studies/references. What are provided in the literature? What other authors found?

Answer to reviewer: We thank the reviewer for this important comment, as guidance in proper exercise is critical. We have added the following text in the discussion: “...highlight a potential benefit for maintaining exercise, particularly under the supervision of a physiotherapist or other sports exercise professional. People with Parkinson’s will benefit from clear guidance on which exercise and physical activities are most beneficial, how often to do them, which is a safe intensity, how many resting periods, which are barriers to expect, etc., [5, 29].”

Comment 4 - Conclusion section

It is suggested to clarify the right initiatives to maintain optimal care? Right initiatives are too general. Can the authors expose some examples for the new approaches to improve healthcare suggested by them in the conclusion section?

Answer to reviewer: We thank the reviewer for the comment and have added the following text in the conclusion to help add on clarity:

“The right initiatives must be implemented to maintain optimal care in the home setting when care cannot be provided directly, such as remote exercise programs or distance guidance exercise for Parkinson's disease supervised by healthcare or exercise professionals.”

Comment 5- Conclusion section

The authors mentioned that “keeping active is not enough to feel benefits in health”, but how did they control the routines and exercises made by people with Parkinson Disease? Were people with Parkinson supervised during the active routines?

Answer to reviewer: We thank the reviewer for this important point. People where asked what type of exercise activities they were doing during the confinement. We have added a reference to the answers given in the results to better clarify what active routines included. We added: Seventeen (63%) had maintained elements of active lifestyle, including informal exercise routines such as daily outdoor walks, home exercises and online sections.”

Reviewer 3 Report

Introduction:

Page 2, line 62: recommend adding the study hypothesis or purpose of the study followed by the question posed by the authors in relation to exercise and Parkinson's.

Methods:

Page 3, line 103: Request authors to add more details on how data was collected on occurrences of falls. Was it falls per week, number of falls in a month, what question was asked to collected this variable and what were the response categories.

 Results:

Line 137: Recommend using the terms 'mean' and 'standard deviation (SD)' instead of 'average' as similar terminology have been used in the descriptive statistic of Methods section (for the purpose of consistency and using statistical language) and results table 1.

Suggestion: Italicize the sub-heading in line 146.

Line 148: please delete the repeated word.

Consider a panel figure for presenting results as all 5 figures present the same box plots representing 2 groups of exercise maintained and interrupted. Also because it makes the results section longer than required.

Page 10, line 270: More precise presentation of data collected on falls is required. 

Table 2: 

The scale PDQ8 states the lower the number the better the health of the individual. In table 2 the Mean is lower for the group 'exercise interrupted' in both ore and post social isolation alluding that interruption of exercise leads to better quality of life in PD patients. Similarly, the scale PDSS has a higher mean for the group 'exercise maintained' in both pre and post social isolation. The scale is defined as higher scores imply greater impairment, based on which the PD patients who maintained exercise experienced greater impairment in quality of sleep. 

Discussion:

These results are important and need to be discussed in detail in the 'discussion' section. The authors have slightly touched upon these but they need to provide additional references and rationale on the suggestions and conclusion one should draw from these results. More explanation is required.  

Author Response

We want to thank the reviewers for their comments and efforts towards improving our manuscript. We have incorporated changes to reflect your suggestions. We have highlighted the changes within the manuscript.

Reviewer 3

Comment 1 - Introduction: Page 2, line 62: recommend adding the study hypothesis or purpose of the study followed by the question posed by the authors in relation to exercise and Parkinson's.

Answer to reviewer: We thank the reviewer for the suggestion. We have added the following text: “Here we aim to answer the question Is being physically active enough or do people with Parkinson need structured supervised exercise? We did this by comparing clinical outcomes from before the pandemic lockdown with the same clinical measures following 2 months of isolation in people with PD to assess differences in motor and non-motor aspects of daily life experiences, quality of life, sleep patterns, falls, in comparison with patients’ perceived worsening after these two months of COVID-19 lockdown according to the type of exercise behavior adopted.”

Comment 2 - Methods:

Page 3, line 103: Request authors to add more details on how data was collected on occurrences of falls. Was it falls per week, number of falls in a month, what question was asked to collected this variable and what were the response categories.

Answer to reviewer: We thank the reviewer. We ask participants to record falls in a monthly paper diary. We analyzed the overall number of falls per participant during the study period. We have added the following text:

“Each participant kept a monthly falls diary. The overall number of falls per participant during the study period was analyzed.”

Comment 3 - Results:

Line 137: Recommend using the terms 'mean' and 'standard deviation (SD)' instead of 'average' as similar terminology have been used in the descriptive statistic of Methods section (for the purpose of consistency and using statistical language) and results table 1.

Answer to reviewer: We thank the reviewer for pointing that out. We had made the change recommended.

Comment 4 - Suggestion: Italicize the sub-heading in line 146.

Answer to reviewer: We thank the reviewer for pointing that out. We had made the change.

Comment 5 - Line 148: please delete the repeated word.

Answer to reviewer: We thank the reviewer for pointing that out.

Comment 6 - Consider a panel figure for presenting results as all 5 figures present the same box plots representing 2 groups of exercise maintained and interrupted. Also because it makes the results section longer than required.

Answer to reviewer: We thank the reviewer for the suggestion. We have changed the presentation form to a panel figure (Figure 1a, 1b, 1c, 1d and 1e) all together (page 7).

Comment 7 - Page 10, line 270: More precise presentation of data collected on falls is required. 

Answer to reviewer: We thank the reviewer for the comment. We have added more information on how falls were collected according to comment above and also have the following text: “Falls occurred in 4 (14%) patients during the 2-month period. Two of these individuals had no history of previous falls. Three of these participants belong to the group that had interrupted exercise habits and one belong to the group who had maintained an active lifestyle.”

Comment 8 and 9 - Table 2: The scale PDQ8 states the lower the number the better the health of the individual. In table 2 the Mean is lower for the group 'exercise interrupted' in both pre and post social isolation alluding that interruption of exercise leads to better quality of life in PD patients. Similarly, the scale PDSS has a higher mean for the group 'exercise maintained' in both pre and post social isolation. The scale is defined as higher scores imply greater impairment, based on which the PD patients who maintained exercise experienced greater impairment in quality of sleep. 

These results are important and need to be discussed in detail in the 'discussion' section. The authors have slightly touched upon these but they need to provide additional references and rationale on the suggestions and conclusion one should draw from these results. More explanation is required.  

Answer to reviewer: We thank the reviewer for the comment. We have added the following text in the discussion:

“Interestingly, the group 'exercise interrupted' in both pre and post-social isolation, perceived to have a better quality of life and lower impairment in quality of sleep than the exercise maintained group. Most exercise intervention research exercise studies [36-38] target people with mild to moderate PD symptoms so little is known on how advanced stages might benefit or perceive benefit. We thus hypothesis that the group that interrupted were mainly constituted with people that were older aged (77.0 to 68.9 years) and worse disease level of disability (H&Y 3 to 2.5), that may not easily perceive worsening and thus not impact quality of life assessments. Living with the disease for longer periods may also reflect a better adjustment to disability and potentially hamper self-assessment due to potential cognitive limitations.”

Round 2

Reviewer 1 Report

The authors met all my comments. 

Author Response

We are thankful for your feedback and contribution to improve our study.

Reviewer 2 Report

I would like to thank you the effort made to answer to the presented questions, and also to follow the suggestions made. I think that the manuscript is clearer and powerful. However, there are few issues that need to be answered to the article prior to publication.

- Can the authors add information to the manuscript about professionals that will beneficiate from these findings?

- Did the authors formulated a hypothesis to this study? Where is it exposed?

Author Response

Reviewer 2

Comment: I would like to thank you the effort made to answer to the presented questions, and also to follow the suggestions made. I think that the manuscript is clearer and powerful. However, there are few issues that need to be answered to the article prior to publication.

  • Can the authors add information to the manuscript about professionals that will beneficiate from these findings?

Answer to reviewer: We thank the reviewer.

In the introduction we add the sentence:

“Thus, people with Parkinson and healthcare and exercise professionals involved in Parkinson's disease management should be able to benefit from information about physical exercise prescription.”

And in the conclusion we have added the following:

“Healthcare (physiotherapist, occupational therapist, others) and exercise professionals that work with people with Parkinson’s and prescribe exercise to patients will benefit from using this data as an example to raise patient awareness on the effect of one type of exercise over the other.

  • Did the authors formulated a hypothesis to this study? Where is it exposed?

Answer to reviewer: We thank the reviewer. We have added the study hypothesis and included it in the introduction section:

Here we aim to answer the question Is being physically active enough or do people with Parkinson need structured exercise? We did this by comparing clinical outcomes from before the pandemic lockdown with the same clinical measures following 2 months of isolation in people with PD to assess differences in motor and non-motor aspects of daily life experiences, quality of life, sleep patterns, falls, in comparison with patients’ perceived worsening after these two months of COVID-19 lockdown according to the type of exercise behaviour adopted.”
